# Insights and Considerations in Development and Performance Evaluation of Generative Adversarial Networks (GANs): What Radiologists Need to Know

**DOI:** 10.3390/diagnostics14161756

**Published:** 2024-08-13

**Authors:** Jeong Taek Yoon, Kyung Mi Lee, Jang-Hoon Oh, Hyug-Gi Kim, Ji Won Jeong

**Affiliations:** 1Department of Radiology, Kyung Hee University Hospital, Kyung Hee University College of Medicine, 23 Kyungheedae-ro, Dongdaemun-gu, Seoul 02447, Republic of Korea; smartyoon7@naver.com (J.T.Y.); khyukgi@gmail.com (H.-G.K.); 2Department of Medicine, Graduate School, Kyung Hee University, 23 Kyungheedae-ro, Dongdaemun-gu, Seoul 02447, Republic of Korea; kaitlynjeong@gmail.com

**Keywords:** artificial intelligence, generative adversarial networks (GANs), medical imaging, high-resolution image synthesis, performance evaluation

## Abstract

The rapid development of deep learning in medical imaging has significantly enhanced the capabilities of artificial intelligence while simultaneously introducing challenges, including the need for vast amounts of training data and the labor-intensive tasks of labeling and segmentation. Generative adversarial networks (GANs) have emerged as a solution, offering synthetic image generation for data augmentation and streamlining medical image processing tasks through models such as cGAN, CycleGAN, and StyleGAN. These innovations not only improve the efficiency of image augmentation, reconstruction, and segmentation, but also pave the way for unsupervised anomaly detection, markedly reducing the reliance on labeled datasets. Our investigation into GANs in medical imaging addresses their varied architectures, the considerations for selecting appropriate GAN models, and the nuances of model training and performance evaluation. This paper aims to provide radiologists who are new to GAN technology with a thorough understanding, guiding them through the practical application and evaluation of GANs in brain imaging with two illustrative examples using CycleGAN and pixel2style2pixel (pSp)-combined StyleGAN. It offers a comprehensive exploration of the transformative potential of GANs in medical imaging research. Ultimately, this paper strives to equip radiologists with the knowledge to effectively utilize GANs, encouraging further research and application within the field.

## 1. Introduction

The incorporation of deep learning across various medical imaging disciplines has notably enhanced the capabilities of artificial intelligence. However, this advancement has also introduced several challenges that warrant attention. First, there is a significant reliance on training data. Medical image analysis often concentrates on images of diseases, leading to an imbalance between normal and diseased image classes. This imbalance, especially with a limited quantity of training data for disease images, can hamper the performance of a model [1]. Second, the process of labeling and segmentation demands considerable time and effort [2]. Third, achieving sophisticated segmentation that meets the high quality required for medical images remains a limitation [3]. To address these challenges, researchers have increasingly turned to generative adversarial networks (GANs) as a solution [4,5,6]. GANs have been employed to create synthetic images of X-rays, computed tomography (CT), and magnetic resonance imaging (MRI), as well as to delineate regions of interest (ROIs) for brain, lung, and liver cancer lesions [4].

Conditional GAN (cGAN), pix2pix, cycle-consistent GAN (CycleGAN), deep convolutional generative adversarial network (DCGAN), progressive growing of GAN (PGGAN), and StyleGAN models have been developed to serve varied objectives in recent research [6,7,8,9,10,11,12,13]. The images generated by these GAN models have been instrumental in data augmentation, creating balanced datasets [5]. They simplify the processing of medical images for tasks such as classification, segmentation, and labeling, not only reducing time and effort, but also ensuring precise segmentation [3,5]. Furthermore, recently developed GANs are being utilized for anomaly detection, which involves identifying abnormal data patterns in an unsupervised manner without extensive label training [14,15,16].

The aim of this paper was to introduce GANs and a cutting-edge artificial intelligence model, and to provide radiologists, particularly those not acquainted with GAN technology, with comprehensive guidelines. We address key considerations in a step-by-step manner, covering the following: (1) an overview of GANs and their variant models, (2) guidance on selecting a GAN model based on this study’s objectives, (3) the training process for images, and (4) evaluating GAN performance. We use examples of CycleGAN and StyleGAN to elucidate key concepts and techniques in each section, facilitating a clearer understanding for the reader. Through this paper, we aim to enhance radiologist comprehension of GANs and encourage further research in the field.

## 2. GAN Architecture, Hierarchy, and Variants

GANs consist of two neural networks designed to generate lifelike images. The generator network aims to produce images that mimic the input image, while the discriminator network works to differentiate between the synthetic and the actual input images. As these networks engage in a competitive dynamic, the GANs progressively create more realistic images (Figure 1A). A key advantage of GANs over conventional convolutional neural networks (CNNs) is their independence from human labeling throughout the image training phase [17]. GANs possess a remarkable ability to analyze the features of unlabeled input images, which allows them to learn without relying on extensively labeled datasets. This breakthrough significantly reduces the reliance on large amounts of labeled data and the significant human labor involved in the traditional deep learning image training methods. Although GANs do not strictly fall under the category of unsupervised learning, their utilization of unlabeled data represents a groundbreaking advancement over existing models [3,5].

### 2.1. Where Do the Generated Images Originate? Noise vs. Image

The generative neural network within GANs synthesizes images through two distinct methods depending on the source material. The first method, known as noise-to-image translation, generates images from random noise. The second method, image-to-image translation, creates images based on existing ones [18]. The application of noise-to-image translation in medical imaging, where anatomical context is essential, presents challenges. To address the limitations associated with generating non-targeted images from noise, cGAN and image-to-image translation techniques have been developed. cGAN enhances the noise-to-image translation process by incorporating conditions such as specific target images or labels into both generative and discriminative networks (Figure 1B) [13]. Image-to-image translation maintains the fundamental structure of the input image domain while modifying detailed characteristics such as appearance, texture, tone, and color (Figure 1C,D). Therefore, image-to-image translation enable the generation of images that accurately represent various disease features while preserving the anatomical shape and location responsible for an organ’s anatomy.

### 2.2. Improvement of Quality for Generated Images

The pix2pix model, built upon cGAN, utilizes image-to-image translation to refine fine details and reduce image blurring (Figure 1C) [12]. It employs a U-net architecture as the generator, which features skip connections between the encoder and decoder to retain detailed information. The discriminator, known as PatchGAN [19], is optimized to highlight high-frequency details. Furthermore, a loss function is introduced to the generator to assess the discrepancy between the input and output images, thereby aiming to enhance the precision of the generated images. CycleGAN addresses a key limitation of the pix2pix model, which requires paired images for learning [11]. The major drawback of medical images is the limited availability of paired diseased and normal images [20,21,22]. CycleGAN is able to learn from unpaired images, allowing it to generate normal images from diseased ones and vice versa (Figure 1D and Figure 2).

Several GAN models enhance resolution through a noise-to-image translation approach, which is distinct from image-to-image translation. DCGAN integrates the structure of CNNs into GANs, which stabilizes the image training process and visualizes some intermediate learning stages [10]. PGGAN incrementally trains the generator and discriminator, starting from a low spatial resolution of 4 × 4 pixels. This method begins with the generation of low-resolution images and gradually introduces new layers to produce higher-resolution details, enabling a stable and swift learning [9]. Developed from PGGAN, StyleGAN is another model capable of creating high-resolution images. While PGGAN has an entangled latent space, making it difficult to change only one of several features, StyleGAN features a disentangled latent space, allowing the modification of individual features while preserving others (Figure 1E) [7,8].

Richardson et al. utilized pixel2style2pixel (pSp) and encoder for editing (e4e) encoders to efficiently encode input images into the latent space of a StyleGAN generator for the creation of human face images [23]. Incorporating an encoder with StyleGAN enhances the ability to modify specific features accurately without altering the original image—a task that is more challenging with StyleGAN alone. While StyleGAN typically generates images through noise-to-image translation, pairing it with an encoder enables image-to-image translation. This method allows for the generation of images tailored to the specific conditions of the input image [23,24]. By combining an encoder, which extracts features from input images, with a GAN that produces high-resolution images, this approach overcomes a previously noted limitation of GANs being restricted to noise-to-image translation. This advancement enables the application of image-to-image translation to medical images requiring high-resolution images, rectifying a previous drawback [25].

## 3. Selecting the Appropriate GAN for the Research Objectives

GANs have been recently utilized for image augmentation, reconstruction, segmentation, and anomaly detection (Table 1) [26]. However, radiologists frequently encounter challenges in selecting the suitable GAN for their specific needs. To aid in research design, we offer three considerations and two examples of applications (Table 2).

### 3.1. Three Considerations

#### 3.1.1. Image-to-Image Translation GANs and Interindividual Anatomic Variance

The first consideration in medical imaging is selecting image-to-image translation GAN variants. Medical images must accurately reflect the basic anatomy of the organ being depicted; thus, image-to-image translation methods are crucial [34,35]. If a noise-to-image translation GAN variant is preferred, it should be combined with an encoder to convert it into an image-to-image translation form for making effective latent space.

However, among GANs utilizing image-to-image translation, there is a variance in the range of structural diversity they can handle, which affects their learning capabilities and performance. The selection of an appropriate GAN should be tailored to the specific organ being generated, considering the degree of interindividual anatomical diversity.

Brain tissue, head and neck areas, chest, solid organs in the upper abdomen, and the musculoskeletal system typically show relatively low variation in anatomical shape and location. For these regions, image-to-image translation methods such as pix2pix, CycleGAN and its variants, or combinations thereof with DCGAN, PRGAN, StyleGAN, and autoencoders, are suitable. However, for generating images of anatomical structures with high variability in location and shape, like the vascular system (including brain vessels) and the bile system, CycleGAN may not perform adequately because of its sensitivity to changes in geometric shapes or positions. Its image generation capability may diminish when there is inconsistency in the distribution or shape of objects across training datasets [11]. Therefore, for research involving images of vessels and the bile duct system, it is recommended to employ high-resolution generators such as DCGAN, PGGAN, and StyleGAN in conjunction with autoencoders [36,37].

#### 3.1.2. High-Quality Image Resolution and Contrast

The second consideration is the resolution and contrast of the images. Noise-to-image translation GAN variants can be used in medical imaging. Many noise-to-image translation GAN variants can implement fine image features. These GAN variants can be used in medical imaging by integrating them with an encoder. Additionally, modifying image-to-image translation GAN variants or combining them with cGANs can improve the fine features of the desired conditions [38].

#### 3.1.3. Unsupervised Detection Models

The third consideration is whether to implement a model capable of unsupervised detection. While some modified image-to-image translation GAN variants or those combined with cGANs can produce high-quality medical images, they may not be suitable for unsupervised anomaly detection. Conversely, noise-to-image translation GAN variants can facilitate unsupervised anomaly detection but may not be optimal for medical imaging. To create a model that meets both high-quality medical imaging and unsupervised detection, a combination of an encoder and noise-to-image translation GAN variants should be considered [39].

Training a GAN combined with an encoder on normal images allows the encoder to extract normal features from an inputted abnormal image within the latent space. If there is a significant discrepancy between the normal features extracted from the latent space and the abnormal features inputted, this difference is recognized as an anomaly and is thus detected. GANs capable of anomaly detection include cGAN, bi-directional GAN (BiGAN), DCGAN, and Wasserstein GAN (WGAN). However, to address the slow pace of iterative optimization, encoder-based architectures have been suggested as enhancements [39,40,41].

Schlegl et al. first developed AnoGAN, which leverages deep convolutional generative adversarial networks (DCGAN) for unsupervised anomaly detection. This innovative method was specifically applied to identify guide markers within optical coherence tomography (OCT) images [42]. After f-AnoGAN [40] and GANomaly [43], GAN models combined with encoders were introduced [44]. Nakao et al. created a model that combines a variational autoencoder (VAE) with α-GAN, aimed at detecting lesions in chest X-rays, such as lung masses, cardiomegaly, pleural effusion, and hilar lymphadenopathy. Despite not achieving the higher performance levels of supervised CAD systems like CheXnet, it achieved an AUROC of 0.7–0.8 for anomaly detection [45]. 

### 3.2. Examples of GAN Application to Brain MR Imaging

Through considering the characteristics and roles of the mentioned GANs, this paper introduces two examples of synthesized images using GANs based on brain MR images from a tertiary hospital in Korea (Appendix A).

#### 3.2.1. CycleGAN—Brain Infarction Images for Augmentation

The first example emphasizes image augmentation, with acute brain infarction being the focal point of interest. Diffusion-weighted images (DWI) were the primary sequence used in this analysis. The CycleGAN model was selected for image augmentation for several reasons: (1) the brain parenchyma’s shape and location tend to be consistent among individuals, (2) CycleGAN has proven effective for augmenting unpaired images, and (3) the hallmark of acute infarction is the alteration in signal intensity that is visible on DWI images. CycleGAN is adept at modifying these specific signal intensity features within normal anatomical regions, all while maintaining the brain’s inherent structure and shape. In this study, CycleGAN employed a U-Net architecture for the generator and PatchGAN for the discriminator (Figure 3). The CycleGAN successfully generated synthetic images of both infarction and normal brain states from the respective normal and infarction images, thereby augmenting high-quality images for acute brain infarction studies (Figure 4).

#### 3.2.2. pSp Encoder-Combined StyleGAN—Brain Vessel Images for Unsupervised Anomaly Detection

The second example focused on unsupervised anomaly detection of cerebral arteries. The target diseases were stenosis, occlusion, and aneurysm, with the maximal intensity projection (MIP) of time-of-flight magnetic resonance angiography (TOF-MRA) serving as the target sequence. The design considerations for this study were multifaceted: (1) an encoder–GAN combination model for anomaly detection, (2) high-resolution image generation was necessary to distinguish between small stenosis and aneurysms, and (3) it was important to avoid GAN models susceptible to structural diversity because of the high anatomical variation of brain vessels. The utilization of StyleGAN combined with a pSp encoder in this study provided a framework capable of accommodating anatomical variance, synthesizing high-resolution images and facilitating unsupervised anomaly detection. This combination was particularly effective in enhancing anomaly detection by altering color and tone without compromising the structural integrity of the vessels (Figure 5). The synthetic TOF-MRA MIP images produced by the StyleGAN and pSp encoder amalgamation successfully depicted not only normal brain vessel imagery, but also lesions, including small aneurysms of approximately 3 mm, mild stenosis, and occlusion (Figure 6 and Figure 7).

## 4. Input Data Training

### 4.1. Image Data Preprocessing Protocols 

The preprocessing of data not only enhances the efficiency of training for synthetic image models, but also ensures that the synthesized images incorporate features that radiologists find valuable [46]. While specific preprocessing protocols for training GANs with medical images have yet to be standardized, established deep-learning preprocessing techniques such as grayscale conversion, normalization, data augmentation, and image standardization can be employed. Data augmentation can include methods like flipping, enhancement, and filtering [46,47].

When utilizing CycleGAN, the chosen preprocessing technique was left–right flip augmentation. This approach aims to augment the image dataset and mitigate the model’s potential bias toward either the left or right side of images. When employing StyleGAN, the preprocessing strategies involved filtering (for vessel segmentation) and color and angle augmentation (for enhancement), as shown in Figure 5. Filtering was instrumental in excluding the background to prevent the model from learning irrelevant areas, such as minor vessels or background noise, which could elevate computational costs and detract from training efficiency. Color and angle augmentations were implemented to minimize the loss compared to the input image by broadening the spectrum of information the model could learn from the image. A common practice in both studies was the standardization of image sizes.

### 4.2. Training Saturation

As with deep learning, it is crucial to ensure that training is adequate while avoiding the inefficiency of superfluous training by monitoring for signs of training saturation. When using the pSp encoder and style-based generator model, it was observed that the one-test loss value progressively decreased as the number of training images increased, eventually reaching a point of saturation (Figure 8). The one-test loss is a value that reflects the cosine similarity between the input and output images within the pSp framework, serving as an indicator of the quality of the test dataset results. A decrease in the one-test loss value was associated with an improvement in result quality. However, when the processed dataset exceeded 170,000 images, the one-test loss value for color-augmented images surpassed 0.9, and no further enhancements in the quality of output images were noted with additional training. Consequently, the training was concluded to minimize the further expenditure of time and resources because of the marginal gains observed.

### 4.3. Performance Improvement and Ablation Study

Every researcher hopes that their deep learning model immediately shows the best performance. However, the performance of deep learning models is influenced by various factors such as the dataset, model architecture, and hyperparameters. Therefore, trial and error are essential in developing a model that achieves optimal performance. Consequently, many researchers have attempted various methods to enhance model performance, such as introducing data augmentation to overcome the limitation of insufficient data. 

An ablation study is also one of the approaches used to improve model performance. It involves surgically removing a part of an organism, such as an organ or tissue, to observe the organism’s overall behavior. This helps to understand the function or role of the removed part. This method is also used in medical image artificial intelligence research. By removing the features of a dataset, the components of a model, design choices, or the modules of a system, researchers can evaluate a model’s performance and understand the role of the removed elements [48,49]. 

Similarly, ablation studies can be applied to GANs (generative adversarial networks) in various ways [50,51,52,53]: (1) to remove components from the generator or discriminator, or to remove specific layers or blocks; (2) to modify the loss function used in GAN training; (3) to alter the input noise vector, or to change the dimensions or distribution of the noise vector fed into the generator; (4) to change training techniques or hyperparameters (learning rate, batch size, or optimizer); (5) to remove control information, or to remove the conditional information, class labels, or additional input information; or (6) to modify the dataset, remove the data of certain classes, or reduce the dataset size.

Ablation studies in GAN development offer benefits. They help identify which components are essential by showing how performance changes when specific parts are removed. This process can optimize a model by eliminating unnecessary elements, making it more efficient [54]. Ablation studies also assist in debugging by identifying problematic components and enhancing the understanding of a model’s workings [55]. These advantages make ablation studies a vital tool for refining and improving GAN models.

## 5. Performance Evaluation

### 5.1. Quantitative Evaluation 

(1)Metrics

In the field of medical imaging, several metrics are commonly employed to quantitatively evaluate GANs, including the Fréchet inception distance (FID), inception score (IS), root mean square error (RMSE), peak signal-to-noise ratio (PSNR), and structural similarity index measure (SSIM) [56]. FID and IS are used to assess the distribution of features between a reference dataset and the synthesized images, providing insights into the quality and diversity of the generated images. In contrast, RMSE and PSNR are metrics that gauge the average differences between the pixels of images [26,36,57]. SSIM has been developed to overcome the limitations of RMSE and PSNR, which may not fully correspond to human visual perception. It evaluates image quality by considering the changes in luminance, contrast, and structural information [58] (Table 3). 

While the SSIM is designed to evaluate image similarity in a way that aligns with human perception, it may not fully capture the level of detail required in the medical field, where fine image features are crucial for accurate diagnoses. A quantitative value that fails to reflect the critical indicators for radiologists, even if deemed satisfactory, could result in an assessment that lacks practical significance [59]. Also, with the pSp encoder and style-based generator model, the values of metrics including SSIM did not provide information that the radiologists could consistently agree upon (Appendix A). 

This principle is particularly relevant in the context of using CycleGAN for DWI, as illustrated in the first example study. When radiologists assess synthetic images, it is essential that the areas of high signal intensity in the DWI correspond accurately to the vessel anatomy depicted in the patient’s actual image. Similarly, a synthetic image should accurately replicate the brain anatomy and signal intensity of the brain parenchyma observed in the actual patient. Despite the numerous reports on GAN models for synthesizing brain images, many studies have overlooked the inclusion of evaluations by radiologists [60,61].

(2)Adversarial evaluation

Another method for quantitatively evaluating synthetic images generated by GANs involves using another GAN or discriminator. This technique has the benefit of offering immediate feedback to the generator, facilitating autonomous improvements in image quality [62]. However, it necessitates a well-trained and compatible GAN or discriminator to be effective. Similarly, this approach may not fully capture the image features that radiologists look for, presenting hurdles for its clinical application (Table 3).

### 5.2. Qualitative Evaluation

When evaluating images synthesized by GANs, it is crucial for clinical practice that radiologists directly examine these images [59,63]. Various methods exist for evaluating synthesized images, including rating, ranking, preference, or pairwise comparison [64]. Evaluators define criteria tailored to their research objectives and categorize the images for assessment.

This approach can overcome the limitations of quantitative evaluations that rely on metrics, as highlighted in many existing studies [32,36,64]. However, it presents challenges such as inconsistencies in the classification and criteria of evaluation items among evaluators, potential bias, and the high demands of time and cost (Table 3). Here, we discuss the methodology and outcomes of qualitative evaluations by radiologists, as is applied in assessing images generated by CycleGAN and StyleGAN. 

(1)Qualitative evaluation of CycleGAN: synthetic normal and acute infarction images on DWI sequences

The performance of the CycleGAN-based image generation model was assessed using a test dataset comprising 30 images. This model was designed to produce images depicting acute infarction lesions, as well as images from which these lesions had been removed. 

A panel of two neuroradiologists, with 10 and 30 years of experience, respectively, evaluated a total of 24 generated images. They were tasked with determining the realism of the generated images in comparison to actual patient images. In instances of diagnostic disagreement, the radiologists engaged in discussion to achieve a consensus. Generated images deemed indistinguishable from real patient images were classified as “consistent.” The image production success rate was determined by dividing the number of images labeled “consistent” by the radiologists by the total number of images in the test dataset without imposing a time limit for this calculation.

The model produced 24 synthetic images from the 30 input images, with half depicting acute infarction lesions and the other half showing these lesions removed (Figure 7 and Figure 8). Out of these 24 generated images, 19 were deemed consistent by the two evaluating radiologists. Thus, from the 30 input images supplied to the model, 19 synthetic images were judged as consistent with real patient images by the radiologists, resulting in an image augmentation productivity rate of 63.3% (Table 4).

(2)Qualitative evaluation of pSp-encoder-combined Style–GAN generator: MIP images of TOF-MRA of normal intracranial arteries

Four radiology residents at a tertiary hospital in Korea were tasked with evaluating normal intracranial arteries using a set of eight synthetic black-and-white images and eight synthetic color images. The criteria for evaluation included the blurring and irregularity of outer margins, the consistency of the vessel diameter, and the ability to distinguish the courses of adjacent or intersecting vessels. The intracranial vessels were categorized into three groups based on their diameter: large vessels, medium vessels, and small vessels. Large vessels comprised the petrous and cavernous segments of the internal carotid artery (ICA); medium vessels included the A1 and M1 segments, supraclinoid ICA, basilar artery (BA), and distal vertebral artery (VA); while small vessels encompassed the A2, M2, and P2 branches.

The evaluation of outer margin blurring and irregularity was assigned scores of 1 for mild, 2 for moderate, and 3 for severe. Mild cases were identified by blurring at the vessel’s outer margin with a smooth contour, moderate cases by blurring with an irregular contour, and severe cases were marked by an irregular contour with lesions potentially interpretable as aneurysms. The assessment of vessel diameter consistency followed a similar scoring system: mild cases displayed minimal diameter irregularity (less than 25% variation), moderate cases had more pronounced irregularity (less than 50% variation), and severe cases exhibited significant irregularity (more than 50% variation) that might suggest moderate or severe stenosis. The separability of the course of adjacent or intersecting vessels was evaluated as 0 for possible and 1 for impossible. It was possible when the courses of adjacent or intersected vessels could be clearly distinguished, but it was impossible when they could not be distinguished (Figure 9).

In the radiological evaluation of intracranial arteries, the overall average score across the evaluators was higher for large vessels than for small vessels across three evaluation criteria. Color-enhanced synthetic images outperformed those with a black background in assessing the separability of vessel courses. Conversely, synthetic images with a black background were superior to color-enhanced images in evaluating the consistency of the vessel diameter. As for the evaluation of blurring and irregularity on the outer margin, color-enhanced images yielded better results for large and medium vessels, whereas black background images were more effective for small vessels (Table 5).

### 5.3. Diagnostic Performance Evaluation

Evaluating radiologists’ diagnostic performance using synthetic images generated by GANs is both time-consuming and labor-intensive. Nonetheless, this method is highly applicable to real clinical settings and ranks as the most reliable among evaluation strategies. Diagnostic performance is analyzed across three categories: using the generated image alone, the generated image alongside an existing image, and the existing image by itself. These categories may further be broken down by disease type. For each, accuracy, sensitivity, and specificity are determined through ROC curves and AUC values (Table 3). The reference standards could be outcomes from biopsies, angiography, or other clinical diagnostic techniques [36].

In a recent study by Jinhao Lyu et al., a computed tomography angiography (CTA)-GAN was employed to create CTA images from abdominal and neck non-contrast images. The diagnostic efficacy of these synthetic CTA images was evaluated using the reference standard of actual CTA images. Diseases were categorized into aneurysms, dissections, and atherosclerosis, with the study reporting an average diagnostic accuracy of 94% on the internal validation set [32].

## 6. Conclusions

GANs have emerged as a powerful tool for medical image analysis, presenting considerable benefits compared to conventional deep-learning techniques. This paper offers a comprehensive overview of GANs, detailing their variations, architecture, and applications within medical imaging, particularly emphasizing their role in radiology. By presenting case studies and experimental results, we demonstrate how to select the appropriate GAN for specific purposes, preprocess the images for training, and evaluate the models in a manner suitable for clinical practice. Our goal is to foster further investigation and utilization of GANs in clinical settings, thereby enhancing patient care through advanced diagnostic tools.

## 7. Future Directions 

There are three key factors in deciding the future of medical imaging with GANs. First, developing models that capture medical features, including anatomy, using effective latent space. Second, using unsupervised techniques to reduce labeling time. Third, evaluating models with clinically relevant qualitative metrics and diagnostic performance rather than just quantitative metrics. Meeting these three conditions for each organ and imaging modality will make GANs the mainstream in future medical AI models.

## Figures and Tables

**Figure 1 diagnostics-14-01756-f001:**
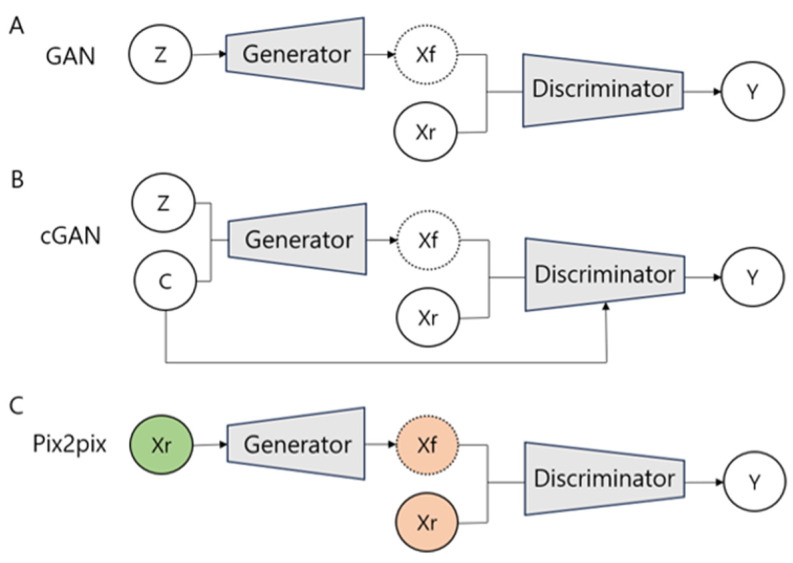
A schematic diagram of various generative adversarial networks (GANs). In a basic GAN structure (**A**), and imposing conditions on both the generator and discriminator in a conditional GAN (cGAN) (**B**). Pix2pix (**C**) uses a paired image set, whereas cycle-consistent GAN (CycleGAN) (**D**) performs image translation between two domains using unpaired images. StyleGAN (**E**) utilizes a mapping network to transform the input Vector Z into an intermediate Vector W before inputting it into the generator.

**Figure 2 diagnostics-14-01756-f002:**
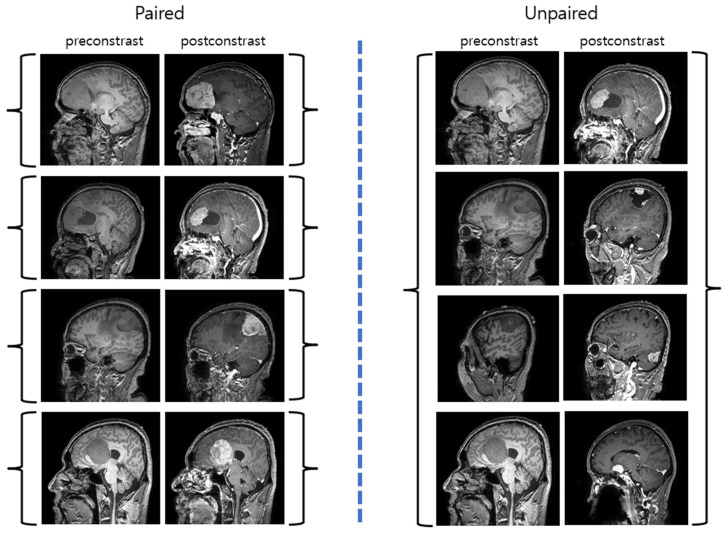
Paired and unpaired image sets. Brain tumor image sets in preconstrast and postcontrast T1-weighted images. Paired image sets are on the **right** and unpaired image sets are on the **left**. Paired images require identical conditions, whereas unpaired images can handle images under different conditions.

**Figure 3 diagnostics-14-01756-f003:**
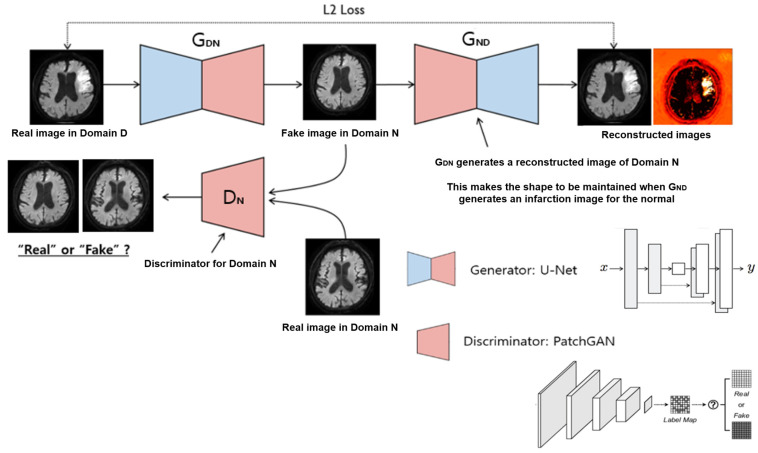
Schematic diagram of a CycleGAN-based generation model. The **top row** shows GENERATOR_DN_, which creates a virtual normal diffusion weighted image (DWI) from a real DWI image of acute infarction. GENERATOR_ND_ creates a virtual acute infarction from a virtual normal image. In the **below** is DISCRIMINATOR_N_, which distinguishes between virtual normal images and real normal images. These two networks compete with each other to make the reconstructed images more realistic and to distinguish between real and reconstructed images, resulting in the generation of high-quality images.

**Figure 4 diagnostics-14-01756-f004:**
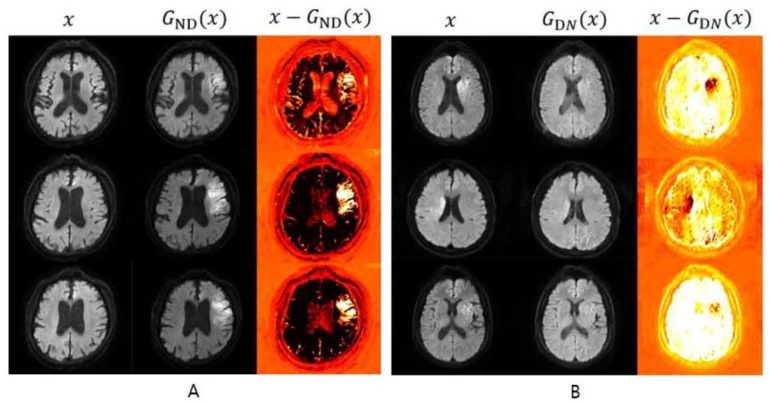
Examples of the synthesized images from the test datasets of GENERTOR_ND_ and GENERTOR_DN_. (**A**) The first column (x) shows the real normal images when given as input. The second column (G_ND_(x)) shows the synthesized acute infarction images. The third column (x − G_ND_(x)) shows the subtraction between the real normal images and the synthesized acute infarction images, which highlights the lesions. (**B**) The first column (x) shows the real normal images when given as input. The second column (G_DN_(x)) shows the synthesized normal images. The third column (x − G_DN_(x)) shows the subtraction between the real acute infarction images and the synthesized normal images, which highlights the lesions.

**Figure 5 diagnostics-14-01756-f005:**
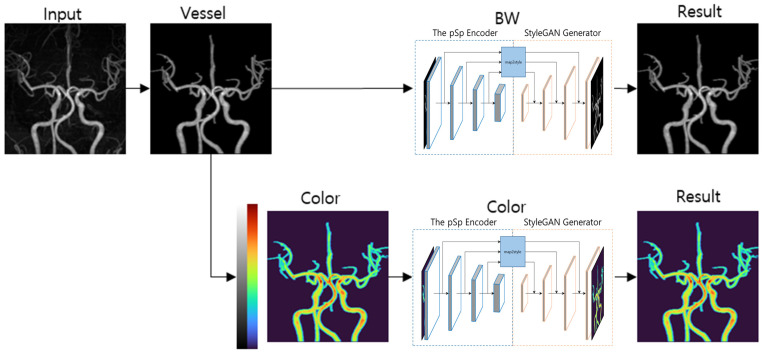
A schematic diagram of the image processing and generation of the pixel2style2pixel (pSp)-encoder-combined StyleGAN generator. Image processing was performed by segmentation of the thick vessels that were relevant to the evaluation following angle augmentation by rotating the image by five degrees each time. Red, green, and blue (RGB) color augmentation of the processed black background image was performed with red for the bright areas and blue for the dark areas. Training each processed black background and the color augmentation images was conducted using a style pSp encoder and GAN generator.

**Figure 6 diagnostics-14-01756-f006:**
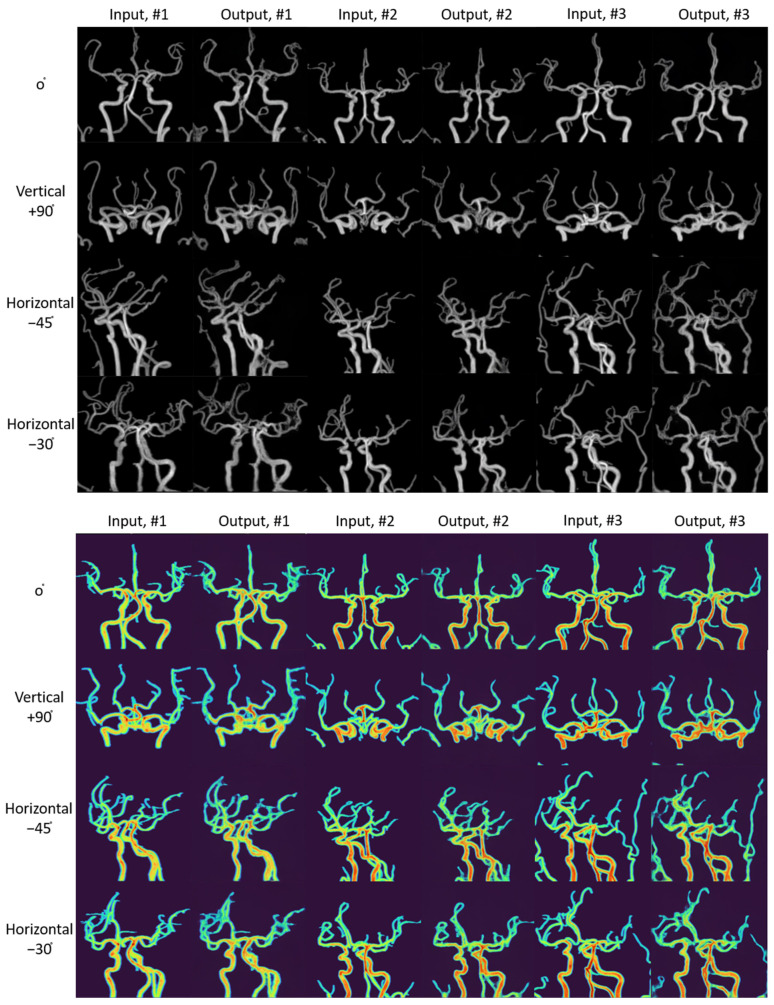
Examples of the StyleGAN generated images from the test datasets of processed time of flight-magnetic resonance angiography (TOF-MRA) maximal intensity projection (MIP) images of normal patients. For three patients of the test datasets, input images and output images were paired and arranged in rows, and vertical +90 degrees, right-rotated horizontal −45 degrees rotation, and horizontal −30 degrees rotation images were arranged in columns.

**Figure 7 diagnostics-14-01756-f007:**
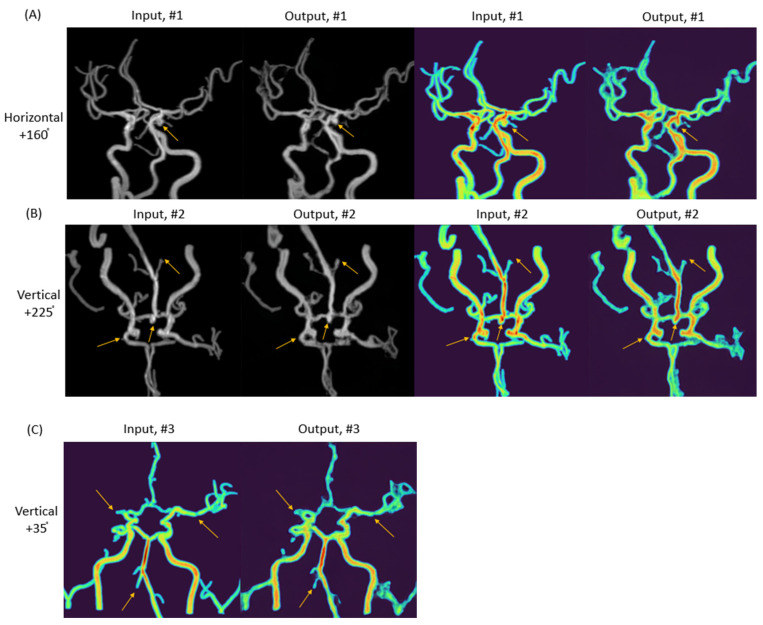
Input and synthetic images of stenosis, occlusion, and aneurysm on StyleGAN. (**A**) Synthetic black background and color augmentation images of 4 mm-sized left posterior communicating artery aneurysm (arrows) at the angle of horizontal 160 degrees and 230 degrees. (**B**) Synthetic black background and color augmentation images of 4 mm-sized basilar top aneurysm and occlusion of right proximal M1 segment and right distal vertebral artery (VA) (arrows) at the angle of vertical 30 degrees. (**C**) Synthetic color augmentation images of stenosis and occlusion of intracranial arteries at the angle of vertical 35 degrees; occlusion of right proximal M1 segment and right distal VA and mild stenosis of left distal M1 segment (arrows).

**Figure 8 diagnostics-14-01756-f008:**
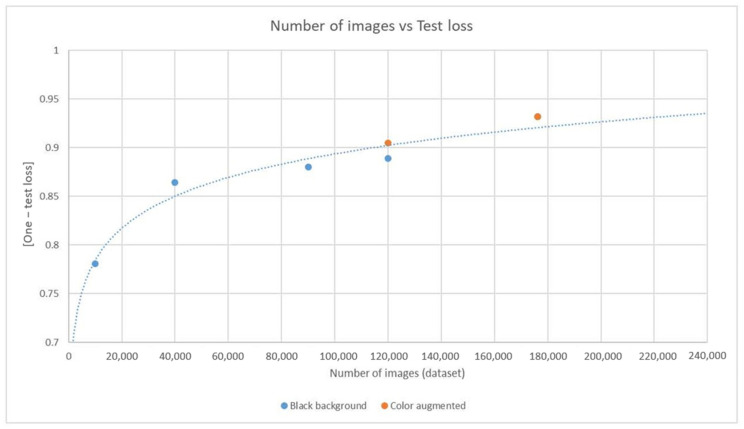
Training saturation of pSp-encoder-combined StyleGAN generator. The cosine similarity between the input and output data embedded in the pSp framework was calculated and expressed as one-test loss. As the number of processed data increased, the one-test loss showed saturation; when the number of color augmentation training data exceeded 170,000, the one-test loss reached 0.93.

**Figure 9 diagnostics-14-01756-f009:**
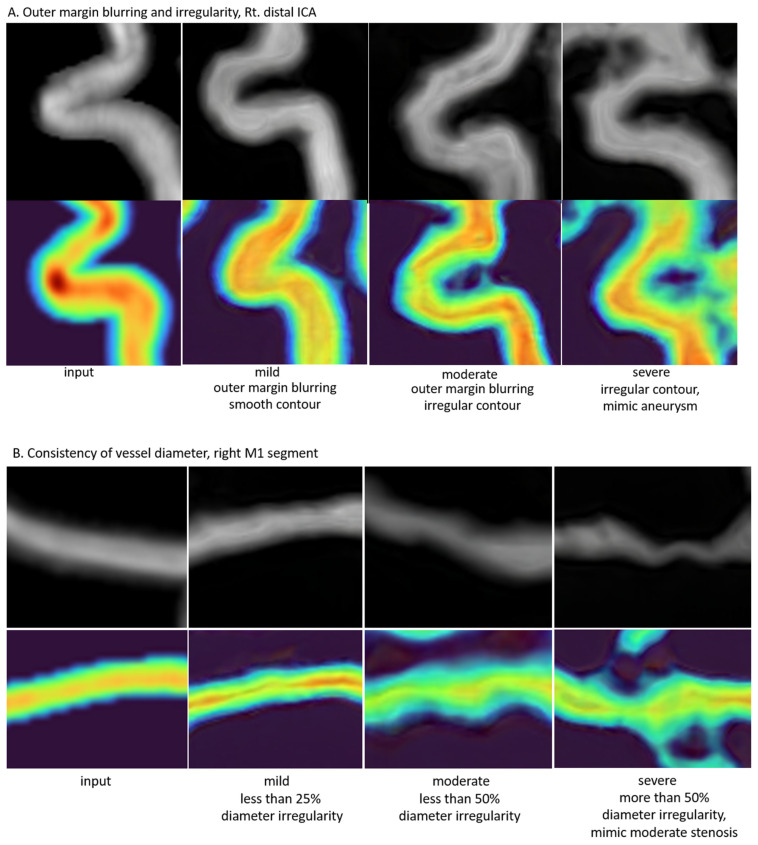
Examples of outer margin blurring/irregularity and consistency of the vessel diameter and separability of the course of adjacent vessels for the qualitative evaluations of radiologists. (**A**) Examples images of outer margin blurring and irregularity in right distal internal carotid artery (ICA), from input to a mild, moderate, and severe degree. (**B**) Examples images of the consistency of the vessel diameter in the right M1 segment, from input to a mild, moderate, and severe degree. (**C**) Examples images of the separability of the course of adjacent or intersected vessels in both ICAs at a horizontal −45 degree, both possible and impossible.

**Table 1 diagnostics-14-01756-t001:** Recent medical image studies using generative adversarial network (GAN) models.

Paper	Target Organ	Modality	Purpose	GAN Variants	Performance Evaluation
Wicaksono et al. (2023) [27]	Intracranial artery	TOF-MRA *	Enhances resolution	Modified pix2pix	MS-SSIM *, 0.87 vs. 0.73 ISSM *, 0.60 vs. 0.35; Improved sensitivity and specificity in detecting aneurysms, stenoses, and occlusions
Mason et al. (2023) [28]	Prostate	mpMRI	Reconstruction	CycleGAN *	Improved quantitative deep learning score No qualitative improvement
Ying et al. (2024) [29]	Liver	CT and MRI	Augmentation	ICycleGAN *	Superior visual quality (SSIM *, PSNR *, NMAE *, FID *)
Yuhan S. and Nak Young C. (2024) [30]	Abdomen	CT → US	Segmentation and reconstruction	S-CycleGAN	Absent suitable metrics and evaluation
Marzieh et al. (2023) [31]	Brain, breast, and blood cancer	MRI Mammography	Unsupervised anomaly detection	f-anoGAN, GANomaly, and multi-KD	Unreliable performance for detecting abnormalities in medical images
Seungjun et al. (2022) [16]	Brain	CT	Unsupervised anomaly detection	CN *-StyleGAN	Shorter post-ADA * triage than pre-ADA * triage by 294 s in an emergency cohort median wait time
Jinhao et al. (2023) [32]	Neck and abdomen	NCCT *	CTA * reconstruction	CTA-GAN	Diagnostic accuracy for vascular diseases (accuracy = 94%)
Wang et al. (2023) [33]	Brain tumor	DSC MRI *	CBV * map reconstruction	Feature-consistent GAN + three-dimensional encoder–decoder network	The highest synthetic performance (SSIM *, 86.29% ± 4.30) Accuracy of grading gliomas (AUC *, 0.857 vs. 0.707)
Seungju et al. (2023) [15]	Breast cancer	Mammography	Unsupervised anomaly detection	StyleGAN2	AUC *, sensitivity, and specificity of the classification performance (70.0%, 78.0%, and 52.0%)

* TOF-MRA = time of flight–magnetic resonance angiography, MS-SSIM = multi-scale structural similarity, ISSM = information theoretic-based statistic similarity measure, CycleGAN = cycle-consistent GAN, ICycleGAN = improved cycle-consistent GAN, SSIM = structural similarity index measure, PSNR = peak signal-to-noise ratio, NMAE = normalized mean absolute error, FID = Fréchet inception distance, CN = closest normal, ADA = anomaly detection algorithm, NCCT = non-contrast computed tomography, CTA = computed tomography angiography, DSC MRI = dynamic susceptibility contrast-enhanced magnetic resonance imaging, CBV = cerebral blood volume and AUC = area under the curve.

**Table 2 diagnostics-14-01756-t002:** The architecture, variants, and characteristics of commonly used generative adversarial networks (GANs).

Architecture		GAN Variants	Detailed Characteristics
Image-to-image translation	Suitable for medical image: preserve anatomical structure	Pix2pix [12]	Applies conditions to generate detailed features Can generate high-quality images Requires paired dataset for training
		CycleGAN * [11]	Uses unpaired data to generate images from different domains Suitable for image augmentation Unsuitable for organs with significant variance in shape and location
		Encoder-combined GANs [23,24]	Generates high-resolution images Enables unsupervised anomaly detection
Noise-to-image translation	Not suitable for the medical images by itself: needs combination with an encoder or other GANs	cGAN * [13]	Assigns conditions to generate required features Needs class labeling Can complicate the training process
		DCGAN * [10]	Generates large-scale and high-quality images Unstable training process
		PGGAN * [9]	Generates high resolution with fine image features Requires high computational costs and a large amount of training data
		StyleGAN [7,8]	Generates high resolution with fine image features Requires high computational costs and a large amount of training data Allows for the selective modification of desired image features

* CycleGAN = cycle-consistent GAN, cGAN = conditional GAN, DCGAN = deep convolutional GAN, and PGGAN = progressive growing of GAN.

**Table 3 diagnostics-14-01756-t003:** Evaluation methods for the GANs’ performance.

Main Categories	Subcategories	Examples	Characteristics
Quantitative	Pixel-level metrics	PSNR *, SSIM *, and RMSE *	An objective evaluation method does not always correlate with radiologists’ evaluations
	Distribution metrics	FID * and IS *	
	Adversarial evaluation	Another discriminator	Immediate feedback to the generator objective evaluation method does not always correlate with radiologists’ evaluations
Qualitative	Radiologist evaluation	Rating, ranking, preference, or pair wise comparison	Applicable to clinical diagnosis, requires time and effort, and there are different standards and biases within the evaluators
Combined	Diagnostic performance	Accuracy, sensitivity, specificity, ROC * curve, and AUC *	Highest reliability in clinical diagnosis requires significant time and effort

* PSNR = peak signal-to-noise ratio, SSIM = structural similarity index, RMSE = root mean square error, FID = Fréchet inception distance, IS = inception score, ROC = receiver operating characteristic, and AUC = area under the curve.

**Table 4 diagnostics-14-01756-t004:** Image generation performance of cycle-consistent GAN (CycleGAN).

	Normal → Synthetic Infarction	Infarction → Synthetic Normal	Total
No. of generated images	12	12	24
No. of consistent images	9	10	19
Image production rate	9/15 (60%)	10/15 (67%)	19/30 (63%)

**Table 5 diagnostics-14-01756-t005:** Qualitative evaluation and details of synthetic StyleGAN images by radiologists.

		Black Background			Color Augmentation		
		Large Vessel	Medium Vessel	Small Vessel	Large Vessel	Medium Vessel	Small Vessel
Rater 1	Outer margin	1.1	1.1	1.3	1.4	1.8	2.1
	Diameter consistency	1.0	1.5	1.5	1.3	1.3	1.9
	Separability	0.1	0.4	1.0	0.3	0.5	0.6
Rater 2	Outer margin	1.4	1.6	1.9	1.3	1.3	2.8
	Diameter consistency	1.5	1.8	2.3	1.9	2.4	2.4
	Separability	1.0	1.0	1.0	1.0	1.0	1.0
Rater 3	Outer margin	1.0	1.1	1.3	1.0	1.1	1.6
	Diameter consistency	1.0	1.1	1.0	1.0	1.1	1.3
	Separability	0.7	0.8	1.0	0.4	0.8	0.8
Rater 4	Outer margin	1.5	1.6	2.1	1.1	1.3	2.0
	Diameter consistency	1.5	1.6	2.6	1.5	1.8	1.9
	Separability	1.0	1.0	1.0	0.4	0.5	0.9
Total Average	Outer margin	1.3	1.4	1.6	1.2	1.3	2.1
	Diameter consistency	1.3	1.5	1.8	1.4	1.6	1.9
	Separability	0.7	0.8	1.0	0.5	0.7	0.8

## Data Availability

The raw data supporting the conclusions of this article will be made available by the authors on request.

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
