# Peer review of "Insights and Considerations in Development and Performance Evaluation of Generative Adversarial Networks (GANs): What Radiologists Need to Know"

_diagnostics, 2024, doi:10.3390/diagnostics14161756_

Round 1
Reviewer 1 Report (Previous Reviewer 1)
Comments and Suggestions for Authors
No new comments
Reviewer 2 Report (Previous Reviewer 2)
Comments and Suggestions for Authors
I have no further comments.
Comments on the Quality of English LanguageN/A
Reviewer 3 Report (Previous Reviewer 3)
Comments and Suggestions for Authors
All the recommendations were performed thank you
This manuscript is a resubmission of an earlier submission. The following is a list of the peer review reports and author responses from that submission.
Round 1
Reviewer 1 Report
Comments and Suggestions for Authors
The paper goal as authors stated is to investigate the using GANs in medical imaging, addresses their varied architectures, and the considerations for selecting appropriate GAN models for Radiologists need.
- row 24: " .......in brain imaging with two illustrative examples." which examples??
- row 40: "To address these challenges, research has increasingly turned to generative adversarial Networks (GANs) as a solution." , it is better to say: To address these challenges, researchers have increasingly turned to generative adversarial Networks (GANs) as a solution [ ][ ][ ].
-from row 45 to 53: need more literature review and more cited references
-Table 1: need to be re-organized, the GAN variants are overlapped with characteristics. Are the characteristics belong to all variants in upper part and then in lower part?
-Also, the selection of appropriate GAN according to research objective ( which is the main goal of this paper) is not clear. Thus, this section (3) need to be re-written and re-organized. Sub-section 3.1, 3.2 and 3.3 are not corelated
- The rest of sections need to be reorganized with sequential and consistent ideas so as not to confuse readers
- The conclusion is very abstracted and not relevant to the research outcome and research
Author Response
Comments 1: row 24: " .......in brain imaging with two illustrative examples." which examples??
Response 1: Thank you for pointing this out. We have now added detailed information about the two illustrative examples.
Comments 2: "To address these challenges, research has increasingly turned to generative adversarial Networks (GANs) as a solution." , it is better to say: To address these challenges, researchers have increasingly turned to generative adversarial Networks (GANs) as a solution [ ][ ][ ].
Response 2: Thank you for the suggestion. We have corrected the sentence accordingly.
Comments 3: from row 45 to 53: need more literature review and more cited references
Response 3: Thank you for your feedback. We have added more references and expanded the literature review in this section.
Comments 4: Table 1: need to be re-organized, the GAN variants are overlapped with characteristics. Are the characteristics belong to all variants in upper part and then in lower part?
Response 4: We have additionally specified the main difference between the upper part and the lower part. The GAN variants in the lower part are difficult to use for medical images on their own. However, when combined with an encoder or other image-to-image translation GAN structures, they acquire characteristics that make them applicable.
Comments 5: Also, the selection of appropriate GAN according to research objective ( which is the main goal of this paper) is not clear. Thus, this section (3) need to be re-written and re-organized. Sub-section 3.1, 3.2 and 3.3 are not correlated.
Response 5: We have reorganized the paragraphs to match the flow of Table 1. We re-categorized the content with headings for better understanding.
Comments 6: The rest of sections need to be reorganized with sequential and consistent ideas so as not to confuse readers.
Response 6: We have removed any potentially confusing sentences in the rest of sections. After revising the problematic sentence, the context remained coherent, so we did not make any major organizational changes to the other sections.
Comments 7: The conclusion is very abstracted and not relevant to the research outcome and research.
Response 7: The research and outcomes in this review paper focus on explaining the process of using GANs rather than focusing on the results. We have added statements about the research to the conclusion to match the purpose of citing it.
Thank you for your comments. We have highlighted your suggestions in red with memo within the manuscript file.
Reviewer 2 Report
Comments and Suggestions for Authors
This paper provides radiologists with a better understanding on Generative Adversarial Networks (GANs). The aim is to guide them through the practical application and evaluation of GANs in brain imaging.
1. The main concern is that the adopted GANs in the work are somewhat out of date. Some of them were published 10 years ago. More recently techniques are recommended to be used.
2. The experiments lack the comparisons results and some ablation studies, as well as the practical usage for radiologists based on the analysis.
3. As the quality is important for medical images, it is suggested to review some relevant works, such as RTN: Reinforced transformer network for coronary CT angiography vessel-level image quality assessment.
4. Please improve the presentation quality of the paper. For example, the figures are blurry.
Comments on the Quality of English Language
N/A
Author Response
Comments 1: The main concern is that the adopted GANs in the work are somewhat out of date. Some of them were published 10 years ago. More recently techniques are recommended to be used.
Response 1: Thank you for the consideration. As a practical example of an application, Cycle GAN, a model introduced in 2017, has been widely used in medical imaging in its modified forms. On the other hand, pSp combined Style GAN, introduced in 2021, has not yet been widely adopted in medical imaging. Table 2 lists models based on the basic structure of GAN, which might give readers the impression that they are outdated. Therefore, additional examples of GANs developed recently are presented in Table 1.
Comments 2: The experiments lack the comparisons results and some ablation studies, as well as the practical usage for radiologists based on the analysis.
Response 2: In the study on pSp combined StyleGAN, we also added research data on how the results vary quantitatively with the addition of color and angle. The two presented studies currently have limitations in their clinical application. We have included these limitations and potential improvements for clinical application in the supplementary materials.
Comments 3: As the quality is important for medical images, it is suggested to review some relevant works, such as RTN: Reinforced transformer network for coronary CT angiography vessel-level image quality assessment.
Response 3: We introduced the background of the two experiments through supplementary materials to explain how they differ from previous research. Also, We added recent researches on Table1. Since we focused on the qualitative evaluation of the research, it is challenging to make a direct comparison of image quality with papers that conducted only quantitative evaluations.
Thank you for your suggestion to include the RTN research example. However, as the RTN study differs from our paper that utilizes GANs, we only used it to understand the intent of your question and did not include a direct comparison with RTN.
Comments 4: Please improve the presentation quality of the paper. For example, the figures are blurry.
Response 4: We apologize for the low resolution of the image files. We have submitted high-resolution versions for all the images included in the manuscript. The blurriness of the intracranial vessels in Figures 6, 7, and 9 is not due to the resolution of the images but rather a characteristic of the vessels generated by the GAN. This is an issue that needs to be improved in future work.
Thank you for your comments. We have highlighted your suggestions in red with memo within the manuscript and supplementary files.
Reviewer 3 Report
Comments and Suggestions for Authors
The article beautifully and thoroughly summarizes a current and highly intriguing technology, with appropriate use of visuals. Current developments are presented along with technical details. With these aspects, I believe the article will attract interest and garner citations. I have a few minor revision suggestions: A "future directions" section should be added at the very end of the article to discuss anticipated developments. The title and keywords contain overlapping words. The keywords section should be revised to include different terms.
Author Response
Comments 1: The title and keywords contain overlapping words. The keywords section should be revised to include different terms.
Response 1: Thank you for your suggestion. We have revised the keywords to include terms that do not overlap with the title.
Comments 2: A "future directions" section should be added at the very end of the article to discuss anticipated developments.
Response 2: Thank you for your suggestion. We have added a "future directions" section to discuss anticipated developments.
Thank you for your comments. We have highlighted your suggestions in red with memo within the manuscript file.
Round 2
Reviewer 1 Report
Comments and Suggestions for Authors
No new comments
Reviewer 2 Report
Comments and Suggestions for Authors
Thanks for the response. However, there still no experimental comparisons for the use of GANs as well as ablation studies, making it lack of convince.
Comments on the Quality of English LanguageN/A